# The Association between Physical Activity and Anxiety in Aging: A Comparative Analysis

**DOI:** 10.3390/healthcare11152164

**Published:** 2023-07-30

**Authors:** Estelio Henrique Martin Dantas, Olivia Andrade Figueira, Alan Andrade Figueira, Anita Höekelmann, Rodrigo Gomes de Souza Vale, Joana Andrade Figueira, Helena Andrade Figueira

**Affiliations:** 1Laboratorio de Biociencias da Motricidade Humana (LABIMH), Programa de Pos-Graduacao em Enfermagem e Biociencias (PPGEnfBio), Universidade Federal do Estado do Rio de Janeiro (UNIRIO), Rio de Janeiro 20270-004, Brazil; estelio_henrique@unit.br (E.H.M.D.); olivia2407@yahoo.com.br (O.A.F.); 2Mestrado em Bioetica, Escola de Medicina e Ciencias da Vida, Pontifícia Universidade Católica do Paraná (PUC-PR), Curitiba 80215-901, Brazil; alan_figueira@hotmail.com (A.A.F.); joanafigueira.riosaude@gmail.com (J.A.F.); 3Seniorenzanz-Zentrum, Institut fur Sportwissenschaft, Otto-von-Guericke Universität, 39106 Magdeburg, Germany; anita.hoekelmann@ovgu.de; 4Departamento de Ciencias da Atividade Fisica (DCAF), Instituto de Educacao Fisica e Desportos (IEFD), Universidade Estácio de Sá (UNESA-RJ), Rio de Janeiro 20071-004, Brazil; rodrigogsvale@gmail.com

**Keywords:** angst, elder, aged, physical fitness, observational study

## Abstract

(1) Background: As the worldwide aging population is growing, there is a need to embrace the role of physical activity in the anxiety of older people. Objectives: To analyze anxiety in older people practitioners and non-practitioners of physical activity; (2) Methods: ample composed of 690 older people of both genders, unselected volunteers, residing in Brazil, present (as participants or observers) in selected street races in the state of Rio de Janeiro, Brazil, between 30 October 2019, and 12 March 2020. An instrument composed of the sociodemographic questionnaire and questions from the Beck Anxiety Inventory, BAI, and the Physical Activity Inventory for older people, Baecke-Old. Design: Ex-post-facto observational analytical descriptive research; (3) Results: The average age of the sample was in the range of 65 to 69y, 74% female, 94% completed high school, 69% living with the family, 84% practicing physical activity. Anxiety levels were 26% (without), 35% (mild), 21% (moderate), and 18% (severe). The comparison of anxiety showed a difference between the groups of active and sedentary elderly. Logistic regression analysis considering anxiety (yes or no; dichotomous variable) and physical activity (yes or no; dichotomous variable) and Odds Ratio were performed to identify possible influences of the independent variables PA, gender, marital status, and education on anxiety. Only physical activity was associated with anxiety. (4) Conclusions: The sample data of this research point to the conclusion that physical activity influences anxiety levels with 98% certainty, and it is suggested that it be enriched in the future with different studies with different designs. The older people practitioners of physical activity with a high level of education presented as 26% without, 35% mild, 21% moderate, and 18% severe anxiety. More active individuals are less likely to develop anxiety.

## 1. Introduction

The burden of disease among older people is high and tends to increase, as one in five people will be over 60 years old, according to projections by the World Health Organization (WHO) [1]. The percentage of older people is increasing worldwide, and “anxiety-depressive” states are emerging health conditions in this population group, increasing both morbidity and mortality [2]. With the growth of the older people population around the world, promoting their health by helping them to achieve a better life, and at the same time reducing the burden on care services, has been the action of academics and professionals around the world, governments, international organizations [3]. Anxiety, the most prevalent psychiatric disorder, is associated with a high burden of illness and is often underrecognized and undertreated in primary care [4]. Prevalence rates of anxiety disorders among older people are up to 15% in community samples and 28% in clinical samples of older adults [5]. Anxiety in late life was for many years the “Cinderella” of psychiatric disorders, often overshadowed by the focus on depression and dementia and receiving little attention in research and clinical domains [6]. Characterized by “excessive and anxious worry, difficult to control, occurring most days” [7], its associated manifestations are those of alertness, surveillance, tension, irritability, non-restorative sleep, and gastrointestinal disorders [7]. Given the high prevalence of anxiety disorder and its associated comorbidities, Aaron Beck composed the internationally validated Beck Anxiety Inventory, BAI, a self-report inventory for measuring the severity of anxiety [8].

Aging is also associated with reduced independence and performance and generally increases the vulnerability of older people, while physical activity benefits healthy aging by increasing psychological stability and physical functioning [9]. Physical activity (PA) has been identified as an essential tool for the prevention and management of multimorbidity among individuals with various health conditions [10]. Although there is an increase in longevity worldwide, this does not guarantee the good health of older people, as each society has its culture and traditions that influence the behavior of older people and, consequently, their aging [11]. The importance and effectiveness of promoting PA have brought optimistic views and suggest increasing awareness and training among health professionals [12]. The scarcity or lack of PA is associated with numerous chronic diseases that encourage the practice of PA [10]. The increase in PA with the reduction of a sedentary lifestyle brings among its benefits the reduction of cellular oxidative stress and inflammation, with improvement in muscle adaptation, mental health, sleep quality, cognitive function, as well as weight loss among other improvements in health condition [13]. PA due to its positive impacts on physical and mental well-being has been described as a miracle drug [12], a promising non-pharmacological method to promote health [14], recommended as one of the non-pharmacological efforts to reduce the anxiety that often occurs in older people [15], and available to all people [14]. To reduce the risks of a sedentary lifestyle, the promotion of PA by health professionals is one of the best investments [13] in terms of cost-effectiveness. In various settings around the world, integrated care programs have been developed that support and promote PA through interprofessional collaboration among health professionals [10].

In its global PA recommendation for healthy aging, the World Health Organization (WHO) states that older people should include “recreational, leisure and occupational, domestic, transportation, games, sports and physical exercises” in a minimum of 150 min per week of moderate aerobic exercise, and for additional benefits gradually double this time and practice weight training at least twice a week [16]. It is essential to meet the WHO minimum requirements for a healthy lifestyle in terms of PA. Over 35% of the world’s population does not meet PA norms, continuing to increase the burden of physical inactivity [12]. To measure physical inactivity and PA in older people, an internationally validated protocol that is easy and quick to apply, the Baecke–Old, a modified Baecke inventory for older people, a well-established screening instrument, is among the most frequently adopted protocols [17]. The current scientific standard [18] includes PA for older people, recreation, occupational, leisure, transportation, home care, and a minimum frequency of 150 min per week of aerobic exercise, and the Baecke–Old questionnaire contemplates this approach of PA; therefore, it includes practitioners of leisure physical activity. PA is especially necessary as people age due to physiological changes involving the neuromuscular, cardiorespiratory, and endocrine systems: fibrillar atrophy, muscle infiltration by adipose and fibrous tissue, insulin resistance status, and decrease in muscle strength/mass and growth hormone [19]. PA is highly correlated with improved well-being and has positive effects not only on physical but mental health, improving overall health and reducing the risk of many negative health outcomes; thus, older adults should be as physically active as their functional ability allows [16]. Considering the impact of PA on healthy aging, with psychological stability and physical function, it is necessary to study the effects of PA on anxiety. The objective of this study is to analyze the presence of anxiety in older people practitioners and non-practitioners of PA. 

## 2. Materials and Methods

### 2.1. Sample and Procedure

The G*Power software version 3.1.9.4 [20] was used to estimate the sample size with the input of information: logistic regression with alpha = 0.05, power = 0.80, OR = 1.3, and Pr Ho = 0.2. The sample size calculated with this information was a total of 568 participants. Considering the possibility of sample loss, the study was developed with a sample ~20% larger than estimated. Thus, the sample was composed of 690 both genders older people, defined as individuals over 60 years (WHO, 2002), unselected volunteers who declared themselves to be residents of Brazil, present (as a participant or observer) in one of the selected street races in the state of Rio de Janeiro, Brazil, from 30 October 2019 to 12 March 2020. The street races visited were: Everybody Goes to Búzios, in Búzios on 11/10/2019; To Run, in Rio Claro, on 11/17/2019; Santa Cecilia Race, in Resende, on 11/17/2019; Run and Walk to Run from Barra Mansa, in Barra Mansa, on 11/30/2019; Serra do Mauá Challenge, in Resende, on 12/8/2019; Paraty Trail Run, in Paraty, on 12/15/2019; Merry Christmas, in Resende, on 12/22/2019; Eclipse Night Run, in Rio de Janeiro, on 01/25/2020; Flowers Run, in Rio de Janeiro, on 01/26/2020; Divas Run Summer, Rio de Janeiro, on 02/16/2020; Barra do Pirai Mountain, in Barra do Pirai, on 08/03/2020; Talk Less and Run More, in Rio de Janeiro, on 08/03/2020. The initial strategy was to look for older people volunteers, both participants and observers, who agreed to participate voluntarily, chosen in a completely random way, without being linked to the convenience of any circumstances that could influence the random selection. The instrument was applied individually via face-to-face interviews, and the older people were asked to answer with their past week in mind. The Study Protocol Flowchart was as follows: participants screened for eligibility → eligible participants consented for the descriptive inquiry analytical observational ex post facto research → instrument was administered individually through face-to-face interviews → two independent statisticians processed data using SPSS 25. 

### 2.2. Inclusion Criteria 

Older people living in Brazil aged 60 or more years volunteered to participate in this research, present (as a participant or observer) in the attended street races, and did not have a problem that prevented them from answering the instrument properly. 

### 2.3. Exclusion Criteria

The exclusion criterion was cognitive impairment, which could hinder their ability to answer the instrument. 

### 2.4. Ethics Approval and Consent to Participate

All participants were informed, and all signed a voluntary consent form to participate in the research. This descriptive inquiry analytical observational *ex post facto* research attending the Helsinki Declaration had been approved by the Research Ethics Board—Federal University of State of Rio de Janeiro—UNIRIO, on 30 October 2019, # 3.670.727, Certificate of Presentation for Ethical Appreciation generated by Plataforma Brasil, # CAAE 11053318.0.0000.5285.

### 2.5. Measures

The instrument composed for this research started with a sociodemographic profile, followed by six selected questions from The Beck Anxiety Inventory, BAI, then five selected questions from Baecke–Old Inventory. From the BAI and Baecke–Old Inventory, essential questions were selected, in order to guarantee that the subject was essentially answered, without, however, the applied questionnaire having to contain a very large number of questions that would make access to a more robust sample impossible. The reason behind choosing these questions, in particular, was that they summarized the perception of anxiety and PA. Presenting the proposed research questionnaire to the Research Ethics Committee of the Federal University of the State of Rio de Janeiro and the doctoral council analyzed the feasibility of the doctoral research; the researcher began by questioning whether there are other methods that can be used to investigate a problem, in addition to the use of a full questionnaire, which will produce the same type of results as the complete questionnaire, underlining that all questionnaires can be prone to bias. Next, the researcher stated that, in her understanding, the most important part of the doctoral research process might be the creation of questions that accurately measure the opinions, experiences, and behaviors of the sample. Creating good measures involves writing good questions and arranging them to form the questionnaire, so the variables of interest in questionnaire-based studies are operationalized through a series of items that constitute a newly developed scale, relying on another previously established. The Research Ethics Committee of the Federal University of the State of Rio de Janeiro and the doctoral council approved the use of this questionnaire as the main instrument of this doctoral research.

The sociodemographic questions were: a. full name encoding; b. gender; c. age range; d. who you live with; e. marital status; f. schooling. The age ranges were 60–64, 65–69, 70–74, 75–79, 80–84, 85–89, and over 90. The Beck Anxiety Inventory, BAI, internationally validated [21], allows grading in levels of anxiety, as defined by Aaron Beck. The Beck Anxiety Inventory, BAI, is a validated, self-administered inventory that measures anxiety symptoms over the past 7 days. Symptoms are graded by the respondent on an ordinal Likert scale ranging from 0 (low) to 3 (high). BAI total score ranges from 0 to 63: 0–10: no; 11–19: mild degree of anxiety; 20–30 moderate; 31–63 severe [21]. Six questions were extracted from the BAI: a. Do you feel irritable; b. Unable to relax; c. Fear; d. Palpitation or racing of the heart; e. Nervous; f. Suffocated. The evaluation of the scores suggested by Beck was weighted, resulting in A. 0–9 = no anxiety; B. 10–11 = mild anxiety; C. 12–15 = moderate anxiety; D. 16–18 = severe anxiety. From the Baecke-Old protocol, used to measure Physical Activity for older people, internationally validated [17], five questions were extracted, each with four response options, on an increasing Likert scale, with scores assigned on a 4-point Likert scale from 0 to 3. The five PA questions selected were: a. Do you do some heavy housework; b. how many steps do you go up in a day; c. what form of transport you use; d. Practicing physical activity (walking, running, Pilates, yoga, fighting, swimming, etc.); e. In leisure time you … Answers to question d were: none = 0, ≤ 30 min/3 day-week = 1, ≤30 min/5 day-week = 2, ≤50 min/3 day-week = 3, ≥50 min/5 day-week = 4, The gradation adopted for PA, in compliance with the WHO guideline for PA for active older people, was that in this question the result was 2 or greater than 2 classified as active, at least 30 min/5 day-week, or 50 min/3 day-week = 3, or even 50 min/5 day-week. 

### 2.6. Statistical Analysis

The double-blind statistical methodology was adopted with the results obtained by the researcher and statisticians both neutral and without prior knowledge of the hypotheses tested, and the older people without access to details of the hypotheses that would lead them to bias; and there was testing for intra and inter errors [22]. The data were processed using SPSS 25. The descriptive analysis provided mean and standard deviation. The Kolmogorov–Smirnov test was used to test the distribution norm, and the Pearson Chi-square test was used to analyze the correlation between the variables. Adjusted data analysis was performed using logistic regression, followed by the Enter method to develop a predictive model for anxiety, and Odds Ratio, with a 95% confidence interval to identify the possible influences of the independent variables PA, gender, marital status, and education on anxiety. Values of *p* < 0.05 were considered statistically significant.

## 3. Results

### 3.1. Sociodemographic Data

The sociodemographic profile of older people is shown in Table 1.

### 3.2. The Level of PA in Older People

Among 690 older people subjects that composed this sample, 582 (84%) were actives, while 108 (16%) were sedentary, characterizing that sample as mostly active. 

### 3.3. The Level of Anxiety in Older People

Figure 1 presents a detailed breakdown of the anxiety levels of the sample. Among the older people participants, anxiety levels were distributed as follows: 26% had no anxiety (level 1), 35% experienced mild anxiety (level 2), 21% reported moderate anxiety (level 3), and 18% had severe anxiety (level 4). Figure 1 illustrates that 26% of the older participants reported minimum anxiety levels, 35% experienced mild anxiety, 21% reported moderate anxiety, and 18% reported severe anxiety.

### 3.4. The Level of Anxiety in Older People

Table 2 shows the percentage of answers to each BAI question by the older people in the sample.

Physical signs of anxiety (palpitation and suffocation) were the lowest complaints, while emotional signs (irritability and fear) were the highest (Table 1). 

### 3.5. The Association between Anxiety and Education, Gender and Living Status

Older people showed a significant association of anxiety with education, gender, and housing status. Table 3 shows the comparison of the average levels of anxiety of the elderly by sociodemographic topics, education, gender, and living situation.

The Chi-square to test associations between sociodemographics and anxiety reported that: anxiety and gender were associated (*p* = 0.02); anxiety and schooling were associated (*p* = 0.027); anxiety and marital status were not associated (*p* = 0.761); anxiety and age group showed no association (*p* = 0.944). The Chi-square test associations between sociodemographic variables and PA reported that PA and age range were associated (*p* = 0.001); PA and education were not associated (*p* = 0.436); PA and marital status showed no association (*p* = 0.211); PA and gender showed no association (*p* = 0.217).

### 3.6. The Anxiety Levels According to the Age Groups 

The statistical comparison of anxiety according to the age groups is presented in Table 4. 

Anova between groups showed the sum of squares between groups 2.518 df = 6 and in groups 755,373 df = 683, Z = 0.380 Sigma = 0.892. Kruskal–Wallis Test for Independent Samples showed Sigma = 0.9161183, confirming the null hypothesis that anxiety is the same between age range categories.

### 3.7. The Association between Anxiety and PA in Older People

The percentage levels of anxiety of active and sedentary older people are presented in Table 5 as follows. 

In the statistical comparison of anxiety between active and sedentary, Pearson’s chi-square Test, the hypothesis test Chi2 = 3.4793, *p* = 0.001, reveals there is a difference between the groups, with the chi-square greater than zero, showing that PA influences anxiety levels. The value of *p* = 0.001 (*p* < 5%) resulted in a level of certainty of ≈99% (Degrees of Freedom = 2, Critical Value = 2.99), informing that the difference found is significant, with the chi-square higher than the critical value, 3.4793 > 2.99, and *p* < 0.05.

Table 6 presents the results of the logistic regression and the Odds Ratio (OR) that were used to analyze whether PA, sex, habitability, marital status, and education are factors that can influence the development of anxiety in the sample of the present study.

Logistic regression analysis (considering anxiety (yes or no; dichotomous variable) and physical activity (yes or no; dichotomous variable)) and Odds Ratio were performed to identify possible adjusted associations. The model did not consider the gender variable as a variable that influences anxiety. The other variables were also not significant. Only physical activity was influential (*p* < 0.05). Only physical activity was associated with anxiety (*p* = 0.004). More active individuals are less likely to develop anxiety.

## 4. Discussion

This research confirms the association between anxiety and PA and shows that 18% of the interviewed older people declared suffering from severe anxiety and 21% from moderate anxiety. The comparison of anxiety among active and sedentary older people showed a high level of certainty, 98%, that PA influences anxiety. The older people in the sample revealed a lower level of anxiety among married people (3.26 versus 5.14), women (3.84 versus 4.9), those living with their families (3.14 versus 5.2), and those with university-level education and above (3.16 versus 5.12). Anxiety and schooling were associated (*p* = 0.027). The study suggests that older people living in their own homes report lower levels of anxiety, family support in one’s own home can improve physical and mental health, and gender is also associated with anxiety [23]. In this research, the vast majority fits into the concept of active, according to the WHO’s definition [16], consistent with a history of marked practice among older Brazilian people [24], where the levels of PA found (67) are close to the findings of this research (84.35%), exceed the observations (73.9%) of a survey carried out in the state of São Paulo [25], and surpass studies that focus only on sports and going to the gym, without considering activities of daily living [26]. 

Anxiety is a natural feeling in living beings on a daily basis, of fear or anguish of surviving, an adaptive attribute to deal with changes that may occur or that are occurring. Anxiety is declared pathological when the condition persists for a sufficiently long period, causes physical disturbances, and results in obstruction of daily activities [27]. If it causes clinically significant distress, impairment of daily activities, or interruption of normal functioning, it becomes clinically relevant, but its detection and diagnosis in old age is hampered by comorbidities and cognitive decline, which tend to occur in aging, contributing to its underdiagnosis in this population [5]. Anxiety in older people should be considered a condition of great importance for public health, as it substantially increases the levels of disability, being more prevalent than depression, with serious consequences for the health of older people [6]. Given the prevalence of anxiety, it is crucial to conduct in-depth studies to improve understanding of its effects on health [5]. Anxiety in older people leads to a greater cardiovascular burden and greater cognitive decline, to increased morbidity and mortality; however, older people tend to minimize the symptoms and attribute them to physical illnesses, making their diagnosis difficult [28]. This research found that the most common anxiety symptoms reported by the older people were emotional (irritability and fear) and the least complaints were physical (palpitation and suffocation). Older adults are more likely to experience anxiety directly and to report particular fearful situations, such as fear of being a burden on their families [5]. Research with more than 5000 seniors at the Mayo Clinic reported that irritability is the strongest symptom of anxiety in older people [29]. Anxiety disorders are frequent and costly in older people and may be part of the phenomenology of late-life depression [5].

Aging acts simultaneously on the social, psychological, biological, environmental, historical, cultural, political, and economic levels, providing varied social representations of aging and older people [11]. The effects of aging, such as functional declines in muscle mass, speed, strength, stability, and firmness, are associated with consequences such as fragility and morbidity and contribute to overall well-being limitations; however, PA can effectively counteract these effects [11]. The way older people perceive their aging process is a subjective experience [3]. The evaluation of the aging experiences of objective indicators, such as physical health, social engagement, and security, should not overlap with the examination of their personal perceptions based on their subjective feelings [3]. The experience of aging varies from person to person, and physical and mental health has an important influence on the perception of aging [23]. Anxiety towards aging influences adaptation to the aging process itself, being a mediating factor in attitudes and behaviors towards older people, as well as in adjusting to one’s own aging process [30]. Despite the relatively high prevalence rates, little is known about the experiences, phenomenology, and evaluation of anxiety in older people [5], and the scientific study of anxiety in old age is still incipient [6]. It is unfortunate that there are still few studies on anxiety in the elderly [31], and more studies are needed [30]. With the increasing number of older people in the general population, anxiety will become a prevalent problem in old age and a major cause of access to health care, resulting in substantial social and individual costs [5]. Achieving an elderly-friendly health system is essential [1], suggesting the development of services that reflect population, social and health characteristics, such as the level of education of the population served and the need for social support [32]. Anxiety is the most common psychological state among elderly people who do not live in their own homes and is associated with depression and cognitive impairment [23]. Depression is the most common comorbidity of anxiety disorders [5]. Objective and perceived social isolation and loneliness are risk factors for the deterioration of social life that can influence anxiety, signaling the need to address the subjective factors of social isolation in a complementary way to PA to improve the mental health of older people [33]. 

Inquiring about anxiety, a study carried out in Zanjan, Iran, with 242 elderly people highlighted the importance of providing resilience training for elderly people [34]. The development of a comprehensive care plan to reduce anxiety in the elderly has been suggested [35], and one such strategy is training in adopting a passive posture towards anxiety, a posture known as ‘freezing’ (i.e., a psychological shutdown), which is part of the fight–flight–freeze system of anxiety, and which is already associated with the aging process, and is often adopted by older adults, who are generally better able to regulate their emotions and employ adaptive strategies effectively [36].

A sedentary lifestyle is associated with higher levels of anxiety, with a strong relationship between vitality and mental health, concluding that PA is a protective factor against anxiety in the elderly, with a correlation between low levels of PA and higher levels of anxiety [17]. Individuals with moderate or low levels of physical fitness have a 23% to 47% greater risk of developing a mental health problem when compared to individuals with high levels of physical fitness [37]. Older people in Pará, Brazil, randomly assigned to control and intervention groups in a six-month resistance exercise program, reported a significant reduction in their level of anxiety after 24 weeks of resistance training [8]. Despite abundant evidence linking PA and mental health, highlighting the importance of sustained PA engagement, the use of PA in mental health treatment, however, remains very limited due to uncertainty surrounding the response to exercise treatment in the face of large individual differences and the various domains of mental health [37]. A cross-sectional survey carried out with 294 elderly people in Saghez, Iran, found a mean and standard deviation for anxiety of 33.63 ± 7.40 [38], while the present study showed a mean and standard deviation of 25.5 ± 15.5. A cross-sectional study with 383 elderly people in Khoy, Iran, showed 41% of elderly people suffering from severe anxiety [39]. The current survey found 26% no anxiety, 35% mild, 21% moderate, and 18% severe. PA is linked to physical and mental health, reducing anxiety [2]. A growing body of work supports the effectiveness of both aerobic and resistance exercise paradigms in the treatment of anxiety [37]. Regular PA reduces feelings of anxiety, according to the conclusion of a survey of elderly people in Sragen [15]. In the treatment and management of mental health conditions, particularly depression and anxiety, aerobic and resistance exercise training shows promise with its behavioral and neurobiological mechanisms that directly link exercise to physical, emotional, psychic, and mental health, creating a cycle that allows predicting the long-term effects of exercise on mental health [37].

While aging is associated with frailty and functional limitations due to its irreversible biological process, and even when associated with a sedentary lifestyle, it has multiple effects of comorbidities that tend to predominate in aging, improving overall health, both mental health and physical health, PA has positive effects on the general well-being of the elderly. The social environment and the type of care can influence the individual’s adaptation to the changes that accompany aging [23].

The practice of PA for 60 min, two or three times a week, increases the feeling of well-being, social relationships, social participation, and leisure [40]. In promoting PA, the core content of the training essentially involves three themes: a. the health benefits of PA, b. health promotion through PA, and c. the change in behavior generated by PA [13]. The World Health Organization has advised that health professionals are crucial to the success of PA promotion [12], aligning with the need for changes in healthcare delivery and policies to match the increased need for healthcare that comes with the increase in the elderly population [1]. Empowering the elderly is one of the main objectives of health promotion programs for older people [3]. AF has consolidated experience in the prevention and non-pharmacological treatment of chronic diseases in health systems, although this path is generally underused by the older people [12]. With changes in general population demographics, anxiety disorders later in life will become a source of increasing personal and societal costs. Policies and programs should encourage inactive older adults to become more active and should provide them with the opportunity and incentive to do so [11].

Disentangling anxiety symptoms in older people from the various factors associated with the aging process is essential to distinguish and address them. Appropriate measures must be taken by the older people themselves and by public policies to improve the practice of promoting integrative health in health environments, encouraging increased physical activity among older people, and reducing the prevalence of anxiety in older people. The findings of this study can encourage the implementation of PA in the older people community, contributing to improving the physical and psychological health of older people. It can also contribute to the development and improvement of policies and programs aimed at older people, as well as to the development of public policies aimed at older people.

To date, few studies have examined anxiety in the elderly, detailing its possible association with PA. A strength of this study is the large sample size, and the authors believe that the statistical power of this research is sufficient to expand the existing body of literature, providing evidence for the association between PA and anxiety and details peculiar to older people. The sample size was estimated in GPower. The n used was greater than expected, and the n was calculated for a comparative study of two groups. This study, however, has significant limitations, such as the cross-sectional design and the partially descriptive nature of the collected data. It is important to recognize, as a limitation of this research, that its participants consisted mainly of active older people with an advanced academic level, unlike other layers of the older people population. About the limitations of the study, pointing out possible selection biases, the research involves the selection bias of its sampling having been carried out in places where physical activity is practiced, therefore potentially having people involved in sports and physical activities. The researchers sought to control this selection bias by including in the sample any person aged 60 or over who was in the geographic area of the street races targeted by this research, either because they live there or because they transit through the area without, however, having any intention of participating in the event. It is suggested that the same research be carried out with selected samples in different areas (hospitals, schools, libraries, and theaters, for example).

## 5. Conclusions

This research confirms the influence of PA on anxiety. The comparison of anxiety between active and sedentary older people revealed that PA influences anxiety levels with 98% certainty. The study suggests that older people living in their own homes report lower levels of anxiety, family support in one’s own home can improve physical and mental health, and gender, education level, and marriage are also associated with anxiety. The most common anxiety symptoms reported by older people were emotional, irritability, and fear. Promoting active aging is both an individual and a health system challenge, whereas the aged voice must be sought in reaching operational definitions for the levels of anxiety and as reporters on the levels of anxiety of their own lives.

Some practical implications derived from this study to the real-world setting are the complexity and heterogeneity of aging justify a multidimensional approach to health care to promote successful aging; anxiety is an inevitable aspect of fast-paced life, and the way it’s managed can have a great impact on mental and physical health; coping strategy resources of the older people, such as education and aspects of health care aimed at successful aging, affect the potential for anxiety, can influence the likelihood of experiencing anxiety; a sedentary lifestyle is associated with higher levels of anxiety; the most common anxiety symptoms reported by older people are the emotional ones (irritability and fear), while the least intense symptoms are physical manifestations of anxiety (palpitations and suffocation).

## Figures and Tables

**Figure 1 healthcare-11-02164-f001:**
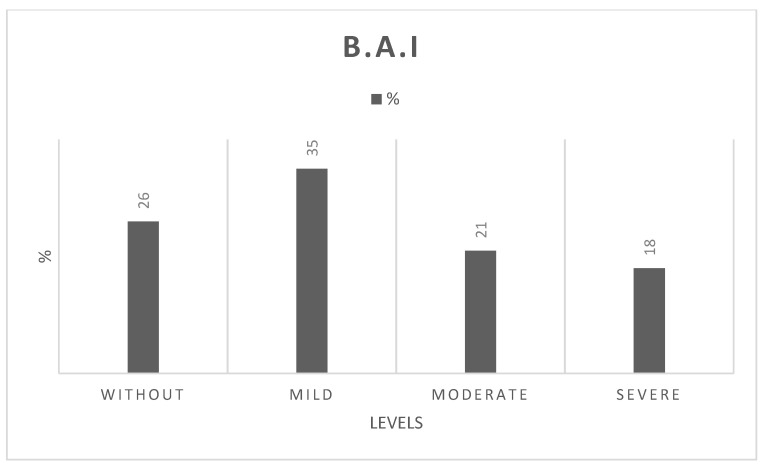
Detailed Level of Anxiety of the Sample. Caption: Without = 0–10; Mild = 11–19; Moderate = 20–30; Severe = 31–63.

**Table 1 healthcare-11-02164-t001:** Sociodemographic Profile.

Questions	Answers	Percent	Absolute
Gender	Feminine	73.6	509
	Masculine	26.2	181
	Other	0.1	1.0
Age-Bracket	60–64	39.4	273
	65–69	32.9	228
	70–74	18.3	127
	75–79	6.1	43
	80–84	2	14
	85–80	0.7	5
	90-	0.6	4
Living Status	Family	68.5	474
	Alone	24.7	171
	Friends	0.7	5
	Others	6.1	42
Marital Status	Single	10.6	73
	Married/Stable Union	54.4	376
	Separated/Divorced	22.8	158
	Widow	12	83
Schooling	Iliterate	0	0
	E.S. Incomplete	2.6	18
	E.S. Complete	1.4	10
	H.S. Incomplete	1.9	13
	H.S. Complete	9.2	64
	U.E. Incomplete	9.2	64
	U.E. Complete	38.4	266
	Pos-Graduation	37	257

Caption: E.S. = Elementary School; H.S. = High School; U.E. = University Education. Source: Figueira et al. [11].

**Table 2 healthcare-11-02164-t002:** Percentage of answers to each BAI question. Caption: irritated = felt irritated in the last two weeks; relax = felt unable to relax; afraid = being afraid; palpitation = feeling heart palpitation; e. nervous = feeling nervous; suffocated = feeling suffocated.

BAI	Irritated	Relax	Afraid	Palpitation	Nervous	Suffocated
No	23.5%	40%	30.4%	50.4%	33.5%	60.3%
Slightly	43.6%	38.5%	39.3%	34.4%	44.4%	27.3%
Moderately	31.3%	21.0%	28.6%	13.8%	20.5%	10.5%
Seriously	1.6%	1.5%	1.7%	1.4%	1.6%	1.9%

**Table 3 healthcare-11-02164-t003:** Average anxiety levels by sociodemographic.

Sociodemographic	Characteristic	Anxiety Level	*p*-Value
Married	Yes	3.26 ± 1.54	
	No	5.14 ± 2.01	
			<0.001
Gender	Male	4.9 ± 2.11	
	Female	3.84 ± 2.1	
			=0.02
Level of education	Below University	5.17 ± 1.85	
	University and up	3.26 ± 1.4	
			=0.027
Living status	Family	3.14 ± 1.3	
	Others	5.2 ± 1.9	
			=0.037
Sample Anxiety	Absolute	4.6 ± 2.8	
Sample Anxiety %	Percentile	25.5 ± 15.5	

**Table 4 healthcare-11-02164-t004:** Anxiety levels by age groups. Caption: Age = Age groups; sd = standard deviation; min = minimum; max = maximum.

Age	Mean	sd	min	max	n
60–64	1.31	1.06	0.00	3.00	271
65–69	1.25	1.06	0.00	3.00	227
70–74	1.30	1.01	0.00	3.00	127
75–79	1.29	1.04	0.00	3.00	42
80–84	1.29	1.14	0.00	3.00	14
85–89	0.80	0.84	0.00	2.00	5
90-	1.75	1.50	0.00	3.00	4

**Table 5 healthcare-11-02164-t005:** Percentage levels of anxiety by PA.

BAI	% Active	% Sedent
No	35.36%	29.29%
Mild	27.99%	20.20%
Moderate	19.34%	27.27%
Severe	17.31%	23.24%

**Table 6 healthcare-11-02164-t006:** Determining factors of anxiety in the present study sample. Caption: Ind. Var. = Independent variables; Gender (1) is a model construct and refers to female and male genders, the gender variable has no subcategories, it is dichotomous; Living Status (1) = Family; Living Status (2) = Alone; Living Status (3) = Friends & Others; Marital Status (1) = Married; Marital Status (2) = Single; Marital Status (3) = Divorced; Marital Status (4) = Widow; Schooling (1) = Iliterate and Elementary School Incomplete and Complete; Schooling (2) = High School Incomplete and Complete; Schooling (3) = University Education; Schooling (4) = Pos-Graduation; B = Beta (the curve angle); Adj. OR = Adjusted OR.

Ind. Variables	B	S.E.	Wald	df	*p*-Value	Adj. OR	95% CI (OR)	
							Lower	Upper
PA	−0.649	0.222	8.517	1	0.004	0.522	0.338	0.808
Age 60–64	0.068	1.030	0.004	1	0.947	1.071	0.142	8.064
Age 65–69	−0.194	1.031	0.035	1	0.851	0.824	0.109	6.214
Age 70–74	−0.205	1.039	0.039	1	0.844	0.815	0.106	6.247
Age 75–79	−0.046	1.074	0.002	1	0.966	0.955	0.116	7.843
Age 80–84	−0.219	1.172	0.035	1	0.852	0.803	0.081	7.992
Age 85-	−1.255	1.545	0.660	1	0.417	0.285	0.014	5.890
Gender (1)	−0.353	0.197	3.198	1	0.074	0.703	0.477	1.035
Living Status (1)	0.306	0.398	0.591	1	0.442	1.358	0.623	2.960
Living Status (2)	0.235	0.353	0.442	1	0.506	1.265	0.633	2.528
Living Status (3)	0.802	0.987	0.659	1	0.417	2.229	0.322	15.439
MaritalStatus (1)	0.567	0.350	2.616	1	0.106	1.762	0.887	3.502
MaritalStatus (2)	0.379	0.304	1.547	1	0.214	1.460	0.804	2.652
MaritalStatus (3)	0.572	0.405	1.997	1	0.158	1.772	0.801	3.916
MaritalStatus (4)	−0.095	0.319	0.089	1	0.765	0.909	0.486	1.699
Schooling (1)	0.604	0.388	2.431	1	0.119	1.830	0.856	3.911
Schooling (2)	0.403	0.335	1.449	1	0.229	1.497	0.776	2.885
Schooling (3)	0.247	0.240	1.066	1	0.302	1.281	0.801	2.048
Schooling (4)	−0.283	0.279	1.029	1	0.310	0.754	0.437	1.301
Constant	−0.440	1.142	0.148	1	0.700	0.644		

## Data Availability

Data are available; a request is required.

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
