# Peer review of "The Association between Physical Activity and Anxiety in Aging: A Comparative Analysis"

_healthcare, 2023, doi:10.3390/healthcare11152164_

Round 1

Reviewer 1 Report (Previous Reviewer 1)

I would like to express my gratitude for the opportunity to review the article titled "A descriptive and analytical observation of the influence of physical activity on the anxiety of older people" authored by Estelio Henrique and colleagues. I appreciate the authors' willingness to incorporate the modifications I previously requested.

Having carefully examined the revised manuscript, I am pleased to acknowledge that all of my recommendations and requests have been diligently considered and appropriately addressed. The authors have demonstrated a commendable level of responsiveness and commitment to improving the quality of their work.

Based on the revisions made, I am confident that the article is now in a suitable state for publication. The authors' efforts in refining the content have significantly enhanced the scientific rigor and validity of their study. The manuscript provides valuable insights into the impact of physical activity on anxiety levels among older individuals, contributing to the existing body of knowledge in this domain.

Once again, I would like to commend the authors for their dedication and meticulousness in addressing the concerns raised during the review process. I strongly recommend the acceptance of this manuscript for publication, as it adheres to the high standards of scientific research and scholarly discourse.

Thank you for your consideration of my assessment, and I remain at your disposal for any further assistance or clarification.

Yours sincerely,

Author Response

Dear reviewer, hope this meets you well.

We are deeply grateful for your thorough analysis of our manuscript.

Based on your suggestions and guidance we incorporated the modifications requested.

We are grateful for your recognition and recommendation for publication, the acceptance of our manuscript for publication, affirming that it adheres to the high standards of scientific research and scholarly discourse.

Thank you for your competent analysis that helped us provide a better way to expose the data, and so on.

Please accept our best wishes

Helena Figueira & coauthors

Reviewer 2 Report (New Reviewer)

Dear Authors,

Thank you for allowing me to review your manuscript titled, "A descriptive and analytical observation of the influence of physical activity on the anxiety of the older people," submitted to the Healthcare Journal. I appreciate the time and effort you have put into your research. However, there are several areas in the manuscript that require significant improvement before it can be considered for publication. My comments, organized based on their occurrence in the manuscript, are as follows:

1. **Title**: The current title of your manuscript, while broadly descriptive of your research, could be more succinct and informative. Here are a couple of suggestions:

   - "Anxiety Profiles in Older Adults: Exploring the Influence of Physical Activity"

   - "The Association between Physical Activity and Anxiety in Aging: A Comparative Analysis"

   Please consider these suggestions or develop a more suitable title based on the essence of your study.

2. **Keywords**: The current keywords included in your manuscript are largely replicated from the title. In scientific publishing, it is advisable to use unique keywords that broadly encompass the article's content. Please revise your keywords using the MeSH (Medical Subject Headings) style for consistency and precision.

3. **Figures and Tables**: The clarity and quality of your figures and tables need significant improvement. They should be clear, well-structured, and of high quality to efficiently communicate the data. I recommend using professional data visualization tools to enhance their quality and comprehensibility, thereby meeting the standards of MDPI Healthcare publications.

4. **Sample Size Calculation**: Your manuscript lacks a section discussing the calculation of the sample size. Including this information is essential for readers to understand the statistical power of your study and the validity of your results. Please include this information.

5. **Study Protocol Flowchart**: I recommend including a flowchart outlining your entire study protocol. This visual representation can help readers more effectively understand the progression and structure of your study.

6. **Statistical Analysis**: The statistical analysis in your research is commendable. The methodology used and the presentation of results are clear and appropriate.

7. **Language**: The level of English proficiency demonstrated in the manuscript needs improvement. It is evident that the manuscript was not written by native speakers, leading to potential misunderstandings due to nuances or subtleties being lost in translation. I found many passages that require significant improvement in academic English. I strongly recommend having your manuscript thoroughly reviewed by a native academic English speaker to ensure language accuracy and clarity.

8. **Conclusion and Practical Implications**: The conclusion of your abstract should be reformulated to include the practical implications of your findings. This will help the readers to understand the real-world applications of your research more clearly.

By addressing these points, I believe the clarity and impact of your manuscript will be significantly enhanced, improving its chances for publication in the Healthcare Journal.

Best regards,

Need much improvement

Author Response

Answer from Authors

Dear Reviewer, hope this meets you well.

We are deeply grateful for your thorough analysis of our manuscript.

Based on your suggestions and guidance we incorporated the modifications requested.

Thank you for your competent analysis that helped us provide a better way to expose the data and so on.

Please accept our best wishes Helena Figueira & coauthors

&&&&&&&&&&&&&&&&&&&&&&&&&&&&&&&&&&&&&&&&&&&&&&&&&&&&&&&&&&&&&&&&&&& &&&&&

Comments and Suggestions for Authors

Dear Authors,

Thank you for allowing me to review your manuscript titled, "A descriptive and analytical observation of the influence of physical activity on the anxiety of the older people," submitted to the Healthcare Journal. I appreciate the time and effort you have put into your research. However, there are several areas in the manuscript that require significant improvement before it can be considered for publication. My comments, organized based on their occurrence in the manuscript, are as follows:

  1. **Title**: The current title of your manuscript, while broadly descriptive of your research, could be more succinct and informative. Here are a couple of suggestions: - "Anxiety Profiles in Older Adults: Exploring the Influence of Physical Activity" - "The Association between Physical Activity and Anxiety in Aging: A Comparative Analysis"  Please consider these suggestions or develop a more suitable title based on the essence of your study. Answer: Done
  2. **Keywords**: The current keywords included in your manuscript are largely replicated from the title. In scientific publishing, it is advisable to use unique keywords that broadly encompass the article's content. Please revise your keywords using the MeSH (Medical Subject Headings) style for consistency and precision. Answer: Done
  3. **Figures and Tables**: The clarity and quality of your figures and tables need significant improvement. They should be clear, well-structured, and of high quality to efficiently communicate the data. I recommend using professional data visualization tools to enhance their quality and comprehensibility, thereby meeting the standards of MDPI Healthcare publications. Answer: Done
  4. **Sample Size Calculation**: Your manuscript lacks a section discussing the calculation of the sample size. Including this information is essential for readers to understand the statistical power of your study and the validity of your results. Please include this information. Answer: Done
  5. **Study Protocol Flowchart**: I recommend including a flowchart outlining your entire study protocol. This visual representation can help readers more effectively understand the progression and structure of your study. Answer: Done.
  6. **Statistical Analysis**: The statistical analysis in your research is commendable. The methodology used and the presentation of results are clear and appropriate. Answer: Thank you
  7. **Language**: The level of English proficiency demonstrated in the manuscript needs improvement. It is evident that the manuscript was not written by native speakers, leading to potential misunderstandings due to nuances or subtleties being lost in translation. I found many passages that require significant improvement in academic English. I strongly recommend having your manuscript thoroughly reviewed by a native academic English speaker to ensure language accuracy and clarity. Answer: Done.
  8. **Conclusion and Practical Implications**: The conclusion of your abstract should be reformulated to include the practical implications of your findings. This will help the readers to understand the real-world applications of your research more clearly. Answer: Done.

By addressing these points, I believe the clarity and impact of your manuscript will be significantly enhanced, improving its chances for publication in the Healthcare Journal. Best regards,

Comments on the Quality of English Language - Need much improvement. Answer: Done.

Round 2

Reviewer 2 Report (New Reviewer)

Review Report for MDPI Health care - "An Observational Analysis of Physical Activity's Influence on Anxiety in Older Individuals"

I'd like to commend the authors for the significant improvements made to this manuscript. The article is well-prepared and ready for publication at its current state.

This manuscript is a resubmission of an earlier submission. The following is a list of the peer review reports and author responses from that submission.

Round 1

Reviewer 1 Report

Summary

1. The current study design does not allow for the validation of the conclusion stating that "Physical activity influences anxiety levels with a 98% certainty level." Hence, it is not appropriate to make claims about the influence or effect of exercise on anxiety based on this type of study.

Introduction

1. No changes suggested.

Material and Methods

1. Could the authors please provide clarification on the basis for sample selection?

2. Depending on the study objective, selecting participants from a specific race may introduce significant bias in the results. How did the authors address this potential bias? Do the authors believe that selecting participants from a single race is representative of the target population?

3. The authors mentioned selecting 6 questions from the BAI questionnaire based on participant descriptions. Could you explain the rationale behind choosing these particular 6 questions? Is the validity of the questionnaire established with only 6 questions? Please provide a justification with a reference.

4. It is recommended to include the information described in point 2.5 as supplementary material.

Results

1. The results are challenging to comprehend overall. Presenting certain data, such as point 3.1, in the form of tables or graphs would enhance clarity.

2. The text pertaining to point 3.3 should be included in the "Material and Methods" section: "The Beck Anxiety Inventory (BAI) is a validated, self-administered questionnaire that assesses anxiety symptoms experienced over the past 7 days. Respondents rate symptoms on an ordinal Likert scale ranging from 0 (low) to 3 (high). The total BAI score ranges from 0 to 63, with scores of 0-10 indicating no anxiety, 11-19 indicating mild anxiety, 20-30 indicating moderate anxiety, and 31-63 indicating severe anxiety."

3. The following paragraphs could be combined: "Figure 1 presents a detailed breakdown of anxiety levels within the sample. Among the older participants, anxiety levels were distributed as follows: 26% had no anxiety (level 1), 35% experienced mild anxiety (level 2), 21% reported moderate anxiety (level 3), and 18% had severe anxiety (level 4)." "Figure 1 illustrates that 26% of the older participants reported minimum anxiety levels, 35% experienced mild anxiety, 21% reported moderate anxiety, and 18% reported severe anxiety."

4. It is necessary to include the child's information in table 1.

5. The authors should conduct an adjusted analysis to determine which variables are associated with anxiety.

Discussion

1. The authors mentioned that one of the study's strengths is the sample size. However, it is suggested that the sample size could be larger to further validate the results. In this regard, do the authors believe that the statistical power is sufficient?

Author Response

Dear reviewer, thank you very much for your attentive and competent analysis of our manuscript. We accept all directions with gratitude. Thank you very much

Reviewer 2 Report

The manuscript analyzed the influence of physical activity on the anxiety of the old people by using the Beck Anxiety Inventory, BAI, and the Physical Activity Inventory for the older people, Baecke-Old. If some minor revisions were made, the manuscript is valuable to be published in the journal.

 1) p.3 ...week = 3, 50min/5day-week = 4, (=> 4.) The gradation adopted, in ...

 2) p.4 3.1. Sociodemographic Data : educational level - 4.8% with a doctorate, 8.8% with a master's degree, 1.7% with medical residency, 21.7% with a postê “graduate degree, 38.4% with a complete university level, 9.2% with an incomplete univerê “sity level and 9.2% with a secondary level. => The term ‘medical residency’ is not clear for education level.

 3) Figure titles “Figure 1. Detailed Level of Anxiety of the Sample.” should be located under the figure. Also. the title of the Y axis should be ‘%’ not ‘n’. Also, It's better to show graph lines (X and Y axes).

 4) p.5 Table 2 : It’s better to place p values in a separate column right side to the ‘Anxiety level’ column.

 5) It’s better to include the anxiety levels according to the age groups.

 6) Because the marital status, gender, education level (and maybe age group) were found to affect the anxiety level (Table 2), the homogeneity of the two activity group (active and sedent) in the aspect of these variables is important. You can test that by chi-suare test for the ‘marital status-activity level’, ‘gender-activity level’, ‘education level-activity level’. If the two groups are not homogeneous, then the confounding should be considered.

Author Response

(The authors gave the same response as above.)

Reviewer 3 Report

Review

A descriptive and analytical observation of the influence of physical activity 

on the anxiety of the older people

The article aims to analyse anxiety in elderly practitioners and non- practitioners of physical activity.

The article includes: introduction, materials and methods (in which the following aspects are addressed: Sample and procedure, Inclusion and exclusion criteria, Ethical approval and consent to participate, measures, statistical analyses), results, discussion and conclusions.

Regarding the research methodology, it considers a sample of 690 elderly people of both sexes, unselected, volunteers, residing in Brazil, present (as participants or observers) in selected street races in the state of Rio de Janeiro, Brazil, between October 30, 2019 and March 12, 2020. Instrument composed of socio-demographic questionnaire and questions from the Beck Anxiety Inventory, BAI and the Physical Activity Inventory for the Elderly, Baecke-Old.

Of the 39 bibliographic references, more than 75% have appeared in the last five years.

The results of the study are thoroughly substantiated; the authors include here Sociodemographic data, PA level in the elderly, Anxiety level in the elderly, Association between anxiety and education, gender and life status, Association between anxiety and PA in the elderly).

The conclusions capture some practical implications derived from the study carried out, in a real context:

As the authors also state, this study has significant limitations, such as the cross-sectional design and the partially descriptive nature of the data collected. It is important to recognize, as a limitation of this research, that its participants consisted mainly of active older people with an advanced academic level, as opposed to other strata of the older population.

We believe that the article can be published in its current form.

Author Response

Dear Reviewer Thank you very much for your thorough revision of our manuscritpt. concerning your guidance below we had included this limitation as per your suggestion.

It is important to recognize, as a limitation of this research, that its participants consisted mainly of active older people with an advanced academic level, as opposed to other strata of the older population.

Answer: Done

Round 2

Reviewer 1 Report

Thank you for submitting a revised version of the article entitled "A descriptive and analytical observation of the influence of physical activity on anxiety in older adults." While you have made progress in addressing several of the questions and suggestions from my previous review, there are still areas that require further improvement in order to enhance the overall quality of your work. Allow me to provide you with some specific recommendations:

1) It is imperative to engage in a thorough discussion regarding the significant limitation posed by the sample size. This limitation must be acknowledged and its potential impact on the study's findings should be carefully examined. Additionally, you may consider exploring alternative strategies, such as conducting a power analysis, to gain a clearer understanding of the statistical validity and generalizability of your results.

2) The potential bias introduced by the selection of participants primarily involved in sports careers warrants a comprehensive discussion. It is essential to acknowledge this limitation and provide a detailed explanation of how it may influence the interpretation of your findings. Consider discussing the implications of this bias on the external validity and generalizability of your study, and explore possible ways to mitigate its impact.

3) The inclusion of adjusted analysis is crucial to address the numerous variables that may be associated with and influence anxiety. It is recommended to incorporate a comprehensive discussion and analysis of the potential confounding factors and their influence on the observed relationships. By conducting adjusted analyses, you can provide a more robust and nuanced understanding of the associations between physical activity and anxiety in older adults.

Author Response

Dear reviewer, thank you very much for your attentive and competent analysis of our manuscript. We accepted all directions with gratitude. Thank you very Much

1) It is imperative to engage in a thorough discussion regarding the significant limitation posed by the sample size. This limitation must be acknowledged and its potential impact on the study's findings should be carefully examined. Additionally, you may consider exploring alternative strategies, such as conducting a power analysis, to gain a clearer understanding of the statistical validity and generalizability of your results.

R.: The sample size was estimated in GPower. The n used was larger than expected. I suggest making the discussion including a paragraph with the limitations of the study and pointing out possible selection biases, ok! The n was calculated for comparative study for 2 groups. But for a logistic regression the n becomes 721.

2) The potential bias introduced by the selection of participants primarily involved in sports careers warrants a comprehensive discussion. It is essential to acknowledge this limitation and provide a detailed explanation of how it may influence the interpretation of your findings. Consider discussing the implications of this bias on the external validity and generalizability of your study, and explore possible ways to mitigate its impact.

R.: Use the same paragraph to justify the selection bias involving people linked to sports and physical activities.

3) The inclusion of adjusted analysis is crucial to address the numerous variables that may be associated with and influence anxiety. It is recommended to incorporate a comprehensive discussion and analysis of the potential confounding factors and their influence on the observed relationships. By conducting adjusted analyses, you can provide a more robust and nuanced understanding of the associations between physical activity and anxiety in older adults.

A.: Logistic regression analysis (considering anxiety (yes or no; dichotomous variable) and physical activity (yes or no; dichotomous variable)) and Odds Ratio were performed to identify possible adjusted associations. Only physical activity was associated with anxiety. More active individuals are less likely to develop anxiety.
